# Identification of Target Chicken Populations by Machine Learning Models Using the Minimum Number of SNPs

**DOI:** 10.3390/ani11010241

**Published:** 2021-01-19

**Authors:** Dongwon Seo, Sunghyun Cho, Prabuddha Manjula, Nuri Choi, Young-Kuk Kim, Yeong Jun Koh, Seung Hwan Lee, Hyung-Yong Kim, Jun Heon Lee

**Affiliations:** 1Division of Animal and Dairy Science, Chungnam National University, Daejeon 34134, Korea; seotuna@cnu.ac.kr (D.S.); cshcshh@cnu.ac.kr (S.C.); prabuddhamanjula@yahoo.com (P.M.); slee46@cnu.ac.kr (S.H.L.); 2Bio-AI Convergence Research Center, Chungnam National University, Daejeon 34134, Korea; ykim@cnu.ac.kr (Y.-K.K.); yjkoh@cnu.ac.kr (Y.J.K.); 3SELS Center, Division of Biotechnology, Advanced Institute of Environment and Bioscience, Chonbuk National University, Iksan 54596, Korea; nuri_23@naver.com; 4Department of Computer Science and Engineering, Chungnam National University, Daejeon 34134, Korea; 5Insilicogen Inc., Yongin 16954, Korea

**Keywords:** single nucleotide polymorphism (SNP), principal component analysis (PCA), genome-wide association study (GWAS), linkage disequilibrium (LD), machine learning

## Abstract

**Simple Summary:**

Classifying a target population at the genetic level can provide important information for the preservation and commercial use of a breed. In this study, the minimum number of markers was used in combination, to distinguish target populations based on high-density single nucleotide polymorphism (SNP) array data. Subsequently, a genome-wide association study for filtering target-population-specific SNPs between the case and control groups and principal component analysis with machine learning algorithms could be used to explore various combinations with the minimum number of markers. In addition, the optimal combination of SNP markers was able to produce stable results for the target population in verification studies, in which samples were analyzed.

**Abstract:**

A marker combination capable of classifying a specific chicken population could improve commercial value by increasing consumer confidence with respect to the origin of the population. This would facilitate the protection of native genetic resources in the market of each country. In this study, a total of 283 samples from 20 lines, which consisted of Korean native chickens, commercial native chickens, and commercial broilers with a layer population, were analyzed to determine the optimal marker combination comprising the minimum number of markers, using a 600 k high-density single nucleotide polymorphism (SNP) array. Machine learning algorithms, a genome-wide association study (GWAS), linkage disequilibrium (LD) analysis, and principal component analysis (PCA) were used to distinguish a target (case) group for comparison with control chicken groups. In the processing of marker selection, a total of 47,303 SNPs were used for classifying chicken populations; 96 LD-pruned SNPs (50 SNPs per LD block) served as the best marker combination for target chicken classification. Moreover, 36, 44, and 8 SNPs were selected as the minimum numbers of markers by the AdaBoost (AB), Random Forest (RF), and Decision Tree (DT) machine learning classification models, which had accuracy rates of 99.6%, 98.0%, and 97.9%, respectively. The selected marker combinations increased the genetic distance and fixation index (Fst) values between the case and control groups, and they reduced the number of genetic components required, confirming that efficient classification of the groups was possible by using a small number of marker sets. In a verification study including additional chicken breeds and samples (12 lines and 182 samples), the accuracy did not significantly change, and the target chicken group could be clearly distinguished from the other populations. The GWAS, PCA, and machine learning algorithms used in this study can be applied efficiently, to determine the optimal marker combination with the minimum number of markers that can distinguish the target population among a large number of SNP markers.

## 1. Introduction

Chicken is a rich source of protein in the human diet. The consumption of chicken has increased globally due to increased consumer interest in health; consumption is also rising annually in Korea [1]. It has been reported that Koreans consume about 2347 tons of chicken every year, which equates to more than nine chickens per person [2]. There has been a gradual shift in emphasis from price to quality, including taste and functionality, e.g., the presence of bioactive compounds (L-carnitine, carnosine, glutathione, omega-3 polyunsaturated fatty acids, etc.), for meat products. Many chicken breeds with improved quality have been produced, but methods are required to certify them at the genetic level.

Generally, the methods used to identify chicken breeds are based on morphological features, but meat products already on the market cannot be identified by their morphological characteristics. A precise identification method allowing for verification at the genetic level is required. Animal genetic information can be used for the maintenance and improvement of livestock varieties, based on phenotypes and heritable genetic characteristics. The Korean government is currently developing genetic markers that can distinguish cattle, pig, and chicken breeds [3,4,5]. However, these markers are microsatellite (MS) markers, with a high polymorphism of a single allele, but identifying genotypes requires much-specialized personnel to perform polymerase chain reaction (PCR) and fragment analyses [6]. Alternatives are needed to overcome these challenges [7].

Single nucleotide polymorphism (SNP) markers could be an important marker-based verification system, with the potential to replace MS markers. With the release of the draft genome sequence of the chicken in 2004, genome-wide SNPs have become available for various research applications. However, many restrictions have been placed on their use, due to the expense and the fact that the technology required to customize the desired SNPs is highly specialized [8,9]. In addition, the Illumina 60 k SNP array, which has already been developed and commercialized, has limitations when it is applied to native chicken populations, and there are high costs associated with the use of the Affymetrix 600 k SNP array for genotyping [10,11]. It is becoming increasingly easy to create SNP kits to identify and validate chicken lines/strains developed for the quality of their meat, where various platforms can be used to combine SNPs in a similar manner to the kits used to diagnose diseases [12,13].

Selecting SNP markers that can distinguish among livestock breeds is not easy. Unlike MS markers, SNPs are the same in all varieties/breeds, but normally possess one of three genotypes (AA, AB, or BB). The process of reducing the number of SNP markers for a particular population can affect the results. A fast and accurate SNP marker verification system running on an automated platform is needed as an alternative to existing verification systems based on MS markers [14]. Attempts to use a combination of SNP markers with automated platforms for target breed identification are being made for various varieties and in various fields [15,16,17].

Machine learning is a type of artificial intelligence in which algorithms are developed that allow computers to make predictions through learning via training data [18,19]. The aim of machine learning is to make predictions based on complex data structures (e.g., big data) that cannot be made by humans. Pattern analysis can be applied to identify different animal populations, using genomics information and various classification models. Machine learning algorithms are rarely used in genetic diversity studies. However, a recent study compared classification performance, based on high-density SNPs, between support vector machine (SVM) and Random Forest (RF) models [20,21,22,23]. In the classification of populations based on genomic information, previous studies have compared the F statistic, delta statistic, and eigenvalue of principal component analysis (PCA) to assess classification models derived from machine learning algorithms. The resulting SNP marker combinations have confirmed the utility of some common SNPs, but the SNP combinations used tend to differ greatly among machine learning algorithms [23]. However, a previous study reported that the machine learning model achieved a better classification performance when a small number of SNPs preselected on a given basis was used rather than all genomic SNPs [21,24].

The Korean chicken industry developed rapidly after the Korean War and became industrialized. However, most of the native chicken genetic resources were lost during the war, and the remaining populations were maintained only in small breeding families in backyards. They had low productivity and made little contribution to the overall development of the industry [25]. Thus, a project to restore native chicken breeds was launched in the 1990s, by the Korean government and a few private companies that maintained small-scale native chicken populations [26]. However, during the industrialization period, consumers became accustomed to commercial broilers, and the classification methods used to recognize commercial native chickens, such as plumage and shank color classification, are unclear to consumers. A project to develop a new chicken breeding stock was also initiated by the government, and a precise technique for the identification of this new chicken breed at the genetic level was considered the best way to verify the new native chicken breed and to prevent it being unfairly distributed in the market.

The purpose of this study was to identify the minimum number of SNP markers needed to identify and verify a target chicken population from among other populations, using information obtained from a 600 k SNP genotyping array for chicken.

## 2. Materials and Methods

### 2.1. Experimental Animals

Two sets of samples were used in this study. The initial set included a total of 283 samples (from 20 chicken populations; Sample Set 1) that were used for a high-density SNP array analysis. This analysis was performed to identify a combination of SNP markers capable of distinguishing the new chicken breeding stock. Samples in this set were divided into four groups: purebred Korean native chicken (KNC), commercial native chicken, commercial broiler, and commercial layer (Table 1). The purebred KNC population consisted of pure lines of KNC and adapted chicken lines, which had been preserved by the National Institute for Animal Science (NIAS), RDA (Rural Development Administration), Korea [27]. The second group included three commercial native chicken lines, including a founder group (Hanhyup F (HF), Hanhyup H (HH), and Hanhyup Y (HY)) that yielded a target group for breed identification. These three lines were maintained by a private company and used for commercial chicken (CC) production. The third and fourth groups, commercial broiler and layer lines, respectively, were used as comparison groups (Table 1). Sample Set 2 consisted of 12 populations and 182 samples. Additional samples were included from the abovementioned populations, and a new commercial native chicken breed was used for validation of SNP combinations in the initial sample set. The detailed sample information for both sets is given in Table 1.

### 2.2. DNA Extraction

All samples used in this experiment were collected according to guidelines issued by the Institutional Animal Care and Use Committee of Chungnam National University, who approved this study (approval no. CNU-00486). Genomic DNA (gDNA) was extracted from whole blood samples taken from the wing vein of birds, using an EDTA (Ethylenediaminetetraacetic acid)-coated tube, to prevent coagulation. Muscle tissue samples were obtained from chicken meat purchased from a market. The gDNA extraction was performed according to the manufacturer’s protocol, using a PrimePrep™ genomic DNA isolation kit for blood and tissue (GeNetBio, Daejeon, Korea). The quality and concentration of the extracted gDNA were verified with electrophoresis, using 1% agarose gel, and spectroscopic analysis, using a NanoDrop spectrophotometer (Thermo Fisher Scientific, Waltham, MA, USA).

### 2.3. High-Density SNP Genotyping and Quality Control (QC)

High-density SNP genotyping of the entire genome was performed by using an Axiom 600 k SNP genotyping array for chicken (Affymetrix, Santa Clara, CA, USA). A total of 580,954 genotypes were analyzed, and the data were transformed into a binary file format, using PLINK software (version 1.9; http://pngu.mgh.harvard.edu/~purcell/plink/download.shtml). A total of 545,563 SNPs were obtained from the merged common SNPs from the PLINK binary data, and this result was subjected to a QC procedure, with the two main criteria of genotype error (missing rate > 10%; 1126 SNPs removed) and minor allele frequency (<0.01; 27,317 SNPs removed) used for the selection of SNP markers in genetic diversity analyses. After the QC process, 517,120 SNPs from 20 chicken populations were accepted and used for further analysis. The genetic distances in the chicken populations were calculated by using Nei’s equation, and fixation index (Fst) values were estimated. The formulas for these calculations are as follows:$$Nei^{’}s\;GD=-\ln\frac{\sum_{l}\sum_{u}pop1_{u}pop2_{u}}{\sqrt{\left(\sum_{u}pop1_{u}^{2}\right)\left(\sum_{u}pop2_{u}^{2}\right)}}$$
where ***u*** is the total number of alleles, **l** is the total number of loci, ***pop*1** is the allele frequency of population 1, and ***pop*2** is the allele frequency of population 2. This value was calculated by using R software’s “poppr” package [28,29].
$$F_{st}=\frac{expH_{tol}-expH_{sub}}{expH_{tol}}$$
where $expH_{tol}$ is the average total population heterozygosity and $expH_{sub}$ is the average sub-population heterozygosity. Fst values were derived by using Weir and Cockerham’s calculation method, with the “SNPRelate” package in R [30,31].

A population structure analysis was performed, based on a multidimensional scale (MDS) plot and admixture analysis, to identify similarities and differences between the target population and the other chicken populations. The MDS plot obtained with PLINK was used to analyze information on pair-wise genetic distances via a four-dimensional scale [32]. The genetic components of each population were analyzed by using ADMIXTURE software (version 1.3); the distributions of the genetic components of the populations were compared according to the numbers of random common ancestors based on the optimum K value [33]. The results of the two analyses were represented graphically by a scatterplot and bar graph, using R software [34].

### 2.4. Selection of 96 Candidate SNP Markers for Identification of the Target Population

A detailed summary of the process used for the selection of candidate SNP markers distinguishing the new chicken breeding stock (with HH, HF, and HY as the parental lines) is provided in Figure 1. We used two main strategies to select a marker combination that distinguished the new chicken breeding stock. In the first step, SNPs were selected by using the case/control association analysis tool in PLINK 1.9. In this analysis, the new chicken breeding stock was the case group, and the other populations comprised the control group. *p*-values were derived for each SNP [32]. SNPs were mainly identified in the macro-chromosome, indicating marker selection bias. In the second step, population linkage disequilibrium (LD) was analyzed by using the significant SNPs obtained to identify SNP markers that were evenly distributed throughout the entire genome. Three sets of 96 significant SNP marker combinations were thus obtained. The accuracy of the classification was compared among the three scenarios. In the first scenario, SNPs with significantly lower *p*-values in a genome-wide association study (GWAS) association test were selected. In the second scenario, 1 SNP per LD block was selected. In the third scenario, 50 SNPs per LD block were selected. An MDS plot of each scenario was constructed to show the degree of separation of the target group from the other populations. In addition, custom SNP assays were designed for verification, using additional chicken samples: A total of 182 samples from 12 populations were collected and genotyped by a Fluidigm array (Fluidigm, San Francisco, CA, USA) (see Table 1).

### 2.5. Machine Learning Approach for Determining the Combination with the Minimum Number of Markers Required for Breed Identification

The 96 selected SNPs, which were identified with the 600 k SNP genotyping array, were used as the training dataset. The data obtained via a verification study with the Fluidigm assay were used as the test dataset, with the target population identified by using classification algorithms of machine learning techniques. We applied eight models to classify varieties/breeds: Random Forest (RF; maximum Decision Tree coefficient—maximum number of sub-populations was 20), AdaBoost (AB), quadratic discrimination analysis (QDA), naïve Bayes, nearest neighbor classification (nine neighbors), linear discriminant analysis (LDA), and Decision Tree (DT) classification. We used the “carret” machine learning package in R software to build a classification model [35]. The eight machine learning models shared a common taxonomy. In the PCA based on selected marker information, PC1 (principal component 1) (75.8%) and PC2 (10.7%) had the greatest descriptive power and were entered as independent variables, regardless of whether new native chicken stocks were set as dependent variables. The re-sampling method used to fit each model was the “cross-validation” method.
Class ~ PC1 +PC2

Each machine learning model had its own criterion for determining whether the target population was consistently classified [36,37,38,39]. The sensitivity refers to the proportion of positive values that were accurately determined, i.e., the true-positive rate (TPR):$$TPR=\;\frac{TP}{\left(TP+FN\right)}$$

The specificity refers to the proportion of negative values that were accurately determined, i.e., the true-negative rate (TNR):$$TNR=\;\frac{TN}{\left(TN+FP\right)}$$
where *TP* is the number of true-positive outcomes, *TN* is the number of true-negative outcomes, *FN* is the number of false-negative outcomes, and *FP* is the number of false-positive outcomes [40].

## 3. Results

The HH, HF, and HY populations were shown by crossbreeding tests to be the best combinations for producing new chicken breeding stocks (data not shown). We sought the minimum number of marker combinations required to classify these three founder populations, and the great grandparent (GGP), grandparent (GP), and CCs produced through their mating. The overall procedure for this is shown in Figure 1.

### 3.1. Genetic Diversity Analyses to Identify SNP Marker Combinations

To identify the target chicken population among the 20 populations included in this study, genetic clustering was performed. The genetic components of each population were confirmed through the MDS plot and admixture analysis. The MDS plot showed that PC1 and PC2 explained 44.414% of the total variance. The HH and HF founder groups were clustered together, directly under the clusters of commercial broiler groups (Cobb broiler (Cobb), Arbor Acre (Ab), and Ross broiler (Ross); Figure 2A). Hanhyup A (HA) was also close to the Ross and Cobb populations. In contrast, HY was in a separate cluster that included the Hanhyup S (HS), Hanhyup W (HW), Rhode Island Red C (NC), and Rhode Island Red D (ND) populations. The commercial layer populations, Hyline brown (HL) and Lohmann brown (LO), were located in the −0.05 region of PC1. The most central clusters were identified as purebred KNCs from the Red Korean native chicken (NR) and Yellow Korean native chicken (NY) populations, which were clustered with Cornish H (NH) and Cornish S (NS). These are known as the Cornish breed and were located in adjacent regions of the plot. In addition, HZ was confirmed to be a similar breed to NH. The HG and HV breeds formed the most independent cluster among all populations in this study. In the PCA plot shown in Figure 2B, we see that PC1 and PC3 explained 38.864% of the total variance. HH, HF, and HY were distributed in different areas of the plot from the commercial broiler population and formed distinct clusters from the other breeds (Figure 2B).

The results of a genetic distance analysis and the fixation index (Fst), calculated based on the 96 SNPs selected from the 600 k SNP genotyping array, are shown in Figure 3A. The results were consistent with those of the MDS plot (Figure 2). The HH and HF founder populations were genetically close to the commercial broilers of the Cobb (0.086 and 0.096), Ab (0.097 and 0.107), and Ross (0.086 and 0.095) breeds (Figure 3A). Both of these founder populations were related to meat-type chicken breeds, and HZ (0.065) was also close to these populations. The Fst results confirmed that the genotype frequency was the same between the HH and HF founder populations and meat-type chicken breeds (0.138~0.175). The HY population was closest to the HS (0.093) and HW (0.077) clusters, and these breeds were closer to the commercial layer populations of LO (0.126) and HL (0.129) than the other chicken populations. The Fst confirmed that HY shared genotypes with HS and HW (Figure 3B).

The admixture results for the 20 chicken populations were used to compare the genetic components among the groups. The lowest cross-validation (CV) error was found at K = 13 (Figure 4A and Appendix A
Appendix A). HH, HF, and HY, which were used as the founder populations for the new chicken breeding stock, had independent genetic components, although the HH and HF populations also shared some genetic components. It was also confirmed that the Ab, Cobb, Ross, and HZ chicken populations had similar genetic components. Similar to the MDS results, the HL and LO commercial layer populations had the same genetic components; the Rhode Island Red breeds, NC and ND, also shared genetic components (Figure 4A). The founder populations of the new chicken breeding stock (HH, HF, and HY) had different genetic components. The KNCs (NR and NY) shared some genetic components with other chicken breeds, such as the Cornish breeds (NS and NH), shown by their central location in the MDS plot (Figure 2).

### 3.2. GWAS and SNP Selection

A GWAS was performed to identify the founder populations of the new chicken breeding stock and the other chicken populations: A total of 47,303 SNPs were used to distinguish between the populations (Appendix A
Appendix A). These markers represented about 10% of the 600 k SNP genotyping array data and were obtained by applying Bonferroni correction to the GWAS analysis results, in which the p-value for significance was 0.05. LD pruning of the case and control groups was then performed to select an even number of markers from each chromosome to distinguish the target group. The discriminatory power of the marker combinations was compared with the PCA results. It was confirmed that Scenario 3 (SNPset3) could efficiently distinguish between the different chicken groups (Figure 5C). In Scenario 1 (SNPset1), PCA involved selection only of the most significant SNPs in the GWAS. In Scenario 2 (SNPset2), in which 1 SNP per LD block was selected, the case and control groups were not clearly distinguished. It was confirmed that 95.8% of the SNPs in SNPset1, 72.9% of the SNPs in SNPset2, and 38.5% of the SNPs in SNPset3 were distributed in the largest chicken chromosome, GGA1 (Appendix A
Appendix A).

### 3.3. Breed Identification by Machine-Learning Algorithms Using the Minimum Number of SNPs

We used the eight machine learning classification models to classify the founder group based on the 96 selected markers: More than 98.5% of the case and control samples were distinguished in all models (Figure 6). All models except the naïve Bayes one had 100% identification power, in terms of the sensitivity to confirm TPs and specificity to confirm FPs. The AB, DT, and RF algorithms also sought a solution involving the fewest markers: Breed classification was possible with 36 markers for AB (Figure 7A), 44 markers for RF (Figure 7B), and 8 markers for DT (Figure 7C).

### 3.4. Validation Study Using Additional Samples

Additional samples were collected to verify the ability of the selected marker combinations to distinguish the founder population and their offspring. Samples of founder groups, commercial native chickens (Woorimatdaq ver2 commercial chicken (WM_2), Yelim Farm commercial chicken (Yelim K), and Hyunin commercial chicken (HI)), commercial broilers (Ab, Cobb, and Ross), and commercial layers (LO) were collected from the Korean chicken market. To confirm the discrimination ability of CCs produced by crossing with the founder population, CCs from the HH, HF, and HY populations were used for verification (182 samples; Table 1).

The 96 selected markers from SNPset3 were genotyped by a customized Fluidigm assay for the validation study. The case/control association results were almost the same before versus after adding the verification samples. When we selected the minimum number of SNP markers by using the feature-selection function of the machine learning models for the AB, DT, and RF algorithms, the discriminatory power exceeded 99% for all three models, using only the 283 training samples in Sample Set 1. In the verification study, the machine learning algorithm was trained by using Sample Set 1, and the case and control chicken populations were predicted by using Sample Set 2 as the validation sample (Table 1 and Figure 1). The target population was classified with 99.6%, 97.9%, and 98.0% accuracy by the AB, DT, and RF models, respectively (Table 2; Figure 7).

## 4. Discussion

The ability to identify chicken breeds or brands on the market at the genetic level could increase consumer trust. Previously used mtDNA sequence variation and MS markers remain useful to verify breeds. However, establishing an automated verification system for these methods take a long time, and an experienced operator with analytical skills is also required [4,41]. SNP markers provide limited variant information compared to MS markers; however, a combination of several SNPs can provide sufficient information for classification. In addition, the cost of genotyping is continually falling, and customizable SNP genotyping platforms can be used as next-generation verification tools that can respond accurately and quickly to market demands.

However, identifying the minimum number of markers from a high-density SNP array for the identification of a target population is not simple. In previous studies, independent SNPs determined by canonical discriminant analysis (CDA), the delta statistic, the F statistic, and PCA were used for genetic classification, and breed identification, using low-density SNP arrays, has also been demonstrated [21,23,24,42]. In these studies, using a 600 k SNP genotyping array for chicken, three combinations of 96 SNP markers were selected based on the results of a GWAS and LD analysis, where the new chicken breeding stock (with HH, HF, and HY as the founder populations) was the case and the remaining chicken groups were the controls. The feature-selection function was applied to SNPset3 to determine the minimum number of markers required for discrimination of the target group. The machine learning algorithm showed high discriminatory power (Figure 1).

### 4.1. Identification of Target Chicken Population Based on Genetic Components

New chicken breeding stocks produced by three-way crossing require a combination of shared markers that can be used to clearly distinguish them from other chicken populations. Twenty chicken populations were used in this study (Figure 4). Of these, 12 chicken populations (HH and HF, HG and HV, HS and HW, NC and ND, NS and NH, and NR and NY) had a shared origin; therefore, a total of 14 chicken populations were predicted to be independent chicken breeds. The HS, HW, NC, and ND lines all originated from Rhode Island Red [27], and the CC lines also shared part of their genetic components with the former lines. Thirteen genetic components could be used to determine the origins of the chicken populations; it was difficult to discriminate them by using fewer marker genotypes.

The populations to be classified had HH, HF, and HY as their parental lines. It was difficult to distinguish HH, HF, and HY from the other chicken populations by using a limited number of SNP markers. In terms of genetic distance, HH and HF were very close (0.09), but HY was relatively distant from those two breeds on the MDS plot (genetic distances of 0.25 and 0.27, respectively). The HY population was more closely related to the other chicken populations than HH and HF. Therefore, it was difficult to identify a marker shared by all three founder populations. The same approach was used to classify breeds by population-specific alleles, similar to the existing mtDNA and MS marker classification approaches. However, different results were obtained from using the different marker combinations when the verification samples were added (data not shown) because the SNPs extracted from the array were not conserved in each population. On the other hand, mutations in mtDNA or MS markers do not affect the function of genes and are selected based on the mutation occurring from the maternal origin of the population (mtDNA marker), or the specific allele (MS marker) of the population is used as an identification point for classification. It is, therefore, difficult to identify populations with a small number of samples by using population-specific SNPs.

### 4.2. GWAS and LD Analysis for Identification of the Target Population

Classification analysis was performed to overcome the limitations of the population-specific markers mentioned above. The HH, HF, and HY populations were set as the case group, and the remaining 17 populations were set as the control group. The 96 markers selected by the GWAS were strongly related to the case group. The case and control groups tended to form distinct clusters but were not clearly distinguished by using only the GWAS with significant SNPs. Therefore, LD pruning was performed and confirmed that SNPset3, which selected 50 SNPs per LD block, could clearly distinguish the two chicken groups due to the removal of the sharing of LD blocks between markers, or the relationship between adjacent LD blocks.

Regarding SNPset1, individuals with high genetic similarity had a high degree of clustering. Several samples overlapped in the MDS plot. When using SNPset2 and SNPset3, which selected SNPs based on the LD block, the clusters were separated according to their relationships. If the SNP markers were selected based on their p-values in the GWAS, those having a strong correlation with the case group were affected by the LD relationships based on marker distances. It was confirmed that 95.8% of SNPset1 (92 of 96 SNPs) shared 39 LD blocks on GGA1 (Appendix A
Appendix A). Additionally, 70 SNPs in SNPset2, and 37 in SNPset3, were located on GGA1. Using the AdaBoost model, which had excellent discriminatory power, only six SNPs were selected from GGA1. Thus, many SNPs were strongly related to the case group in GGA1 but provided redundant information and probably interfered with the classification of the two groups. Selection of SNPs in the case group based on GWAS analysis could increase the genetic distance between the case and control groups. It was difficult to distinguish the two groups based on the Fst, but it was confirmed that the genetic distance between the case and control groups was significantly increased (Figure 3). The optimum K-value in the admixture analysis decreased from 13 to 2 when the minimum number of markers was used in the AdaBoost model (Figure 4). This result indicates that the final selection of 36 markers provided a high level of explanation for the target group.

In previous studies, methods for selecting the minimum number of high-density SNP markers for breed identification by using the delta statistic and Fst were reported. More than 300 and 591 breed-specific SNPs were selected by Judge et al. (2017) [24] and Kumar et al. (2019) [17], respectively. These relatively large numbers of SNPs were used to form a panel to discriminate among target breeds. Another study sought to identify the minimum number of markers needed for breed identification, using the delta statistic, PCA, and an RF algorithm [21]. Combinations of markers (48- and 96-SNP panels) capable of distinguishing among various cattle breeds were presented; efficient identification was possible with fewer markers than in previous studies.

Our GWAS and LD analysis was not performed to identify markers capable of distinguishing among all of the populations included in the study, but rather to distinguish only the target population from the others. It is therefore difficult to directly compare the results with those of previous studies. Comparing the Fst, the genetic distance, and the genetic component of the research population before and after marker selection, it was confirmed that the changes of genetic distance and genetic composition as calculated by the selected markers were significant for the target population, including the Fst value. The genetic distances were calculated based on allele frequencies, and the results were similar to those obtained by using the delta score. The explanatory power of the principal component in the PCA analysis increased when using case-associated markers. The validation study results remained consistent after adding samples from other populations that were not used for marker selection. The 96 selected markers (SNPset3) well explained the genetic components of the target chicken group.

### 4.3. Machine Learning Algorithms for Classification of the Case and Control Chicken Populations

Machine learning is a supervised learning approach for classifying new observations that can be used to classify bi-class or multi-class data. Machine learning can be used for voice and handwriting analysis, and document classification. In recent years, machine learning and deep learning algorithms have been used to determine phenotypic associations (e.g., in the genome, transcriptome, and methylome) in “omics” research, and to establish classification models [43,44].

In this study, eight machine learning classification models were used to efficiently identify target chicken populations. PCA was conducted with a machine learning algorithm to confirm whether case and control groups could be distinguished, based on the 96 markers in the SNPset3. All classification algorithms showed 100% breed classification accuracy except the naïve Bayes model (98.5%; Figure 6). The AB, RF, and DT algorithms select a subset of variables through a feature-selection process. In general, machine learning models utilize this method to do the following: (1) simplify the model for easier interpretation, (2) shorten the training time, (3) avoid the dimensional curse problem, and (4) reduce overfitting (i.e., reduce variance) [45]. In this manner, duplicate or less relevant variables are removed, so the minimal number of SNP markers required for efficiently classifying chicken populations can be identified.

With use of feature selection, 36, 44, and 8 SNP markers were selected by the AB, RF, and DT models, which had classification accuracies of 99.6%, 97.9%, and 98.0%, respectively (Table 2). Thus, the target group could be classified by using a small number of markers.

In the validation study including additional samples, both founder group and non-founder group chickens could be classified. The added samples included PL and CC samples from the founder population, and various samples obtained from the Korean chicken market (including new breeds not included in the 600 k SNP genotyping array, e.g., WM_2, Yelim K, and HI). The discriminatory power was excellent, even when samples from a group of chickens with a completely different genetic composition to that used in the initial marker selection process were added (Figure 7). Our SNP markers were not population-specific. Therefore, by adding samples, the allele frequency and breed classification results could change. However, no significant changes were seen, and the cases and control groups were classified with high accuracy (Table 3).

The AB and RF models showed similar clustering results with and without the additional verification samples, while the DT model produced relatively diffuse clusters. Overall, the accuracy was similar among the classification models. The SNP marker combinations selected by the three models can be used for classifying the target chicken population. However, the DT model showed changes in clustering with additional samples, so it requires further verification.

This approach can provide useful information for the development of the best SNP marker combination for use in forensic science, conservation genetics, and livestock traceability systems [46,47,48]. There is scope to further develop our research; for example, Biscarini et al. (2015) [49] presented a model for predicting the root vigor class in sugar beets, with nearly 100% accuracy based on a minimum set of 30–50 SNPs. In this study, they selected the smallest combination of markers required to efficiently predict binary traits. The method of distinguishing populations with combinations of SNP markers can also be applied to explore markers associated with features in groups whose genetic structure has changed due to differences in SNP feature importance. Moreover, machine learning and deep learning methods can classify multi-class groups based on complex types of multi-omics data, and could therefore be further developed to determine the smallest marker combination that can distinguish among many different groups at the same time, beyond the marker combination that separates two groups. In addition, these methods can efficiently distinguish various types of groups and populations, and it could be used to monitor genetic diversity, as well as to protect the right to use certain breeds in the international community, where awareness of breed sovereignty is growing.

## 5. Conclusions

A genetic marker capable of distinguishing among breeds, at the genetic level, is required to protect intellectual property rights and ensure consumer confidence, but the development of conventional mtDNA and MS markers requires large amounts of time and money, as well as expertise. A marker combination with the minimum number of SNPs required for distinguishing the target chicken population could be used to overcome these shortcomings. In this study, the minimum number of SNPs that could identify target chicken populations was determined by using their LD relationship, case/control GWAS, PCA, and machine learning algorithms. As a result, these methods increased Fst and genetic distance values for the selected marker combinations, when comparing target populations to other populations, thereby increasing the identification power. In addition, the feature selection of machine learning models suggested the most effective marker combinations by minimizing redundant marker information. The SNP selection methods used in this study to distinguish target populations at the genetic level can be used to efficiently select a minimal number of genetic markers. These results can be applied to a variety of livestock, as well as chicken populations, and will also be useful in the field of conservation genetics.

## Figures and Tables

**Figure 1 animals-11-00241-f001:**
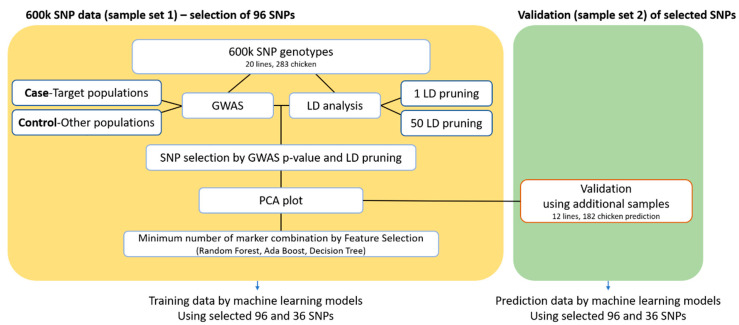
Workflow of the process applied to determine the marker combination with the minimum number of markers required for target population identification. In the validation step, a machine learning algorithm was applied to use Sample Set 1 (283) as training data and Sample Set 2 (182) as prediction data. SNP, single nucleotide polymorphism; GWAS, genome-wide association study for the case/control population; LD, linkage disequilibrium; 1 LD pruning, 1 SNP selected per 1 LD block; 50 LD pruning, 1 SNP selected per 50 LD blocks; PCA, principal component analysis.

**Figure 2 animals-11-00241-f002:**
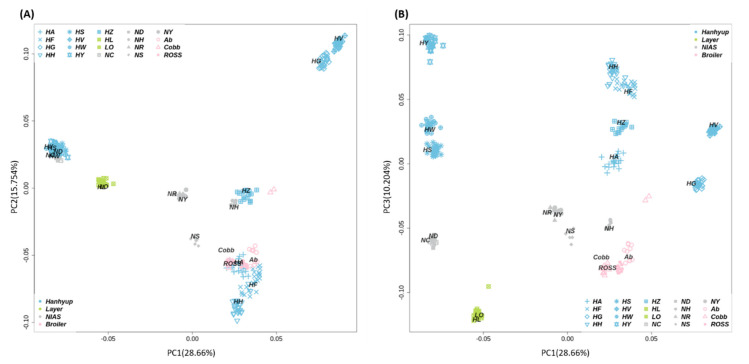
Multi-dimensional scaling (MDS) plots based on 600k SNP genotype data. (**A**) An explanatory power of 44.414% was achieved by using the PC1 (principal component 1) and PC2 dimensions, and (**B**) that of 38.864%, using the PC1 and PC3 dimensions. NC, Rhode Island Red C; ND, Rhode Island Red D; NH, Cornish H; NS, Cornish S; NR, Red Korean native chicken; NY, Yellow Korean native chicken; HH, Hanhyup H; HF, Hanhyup F; HY, Hanhyup Y; HW, Hanhyup W; HS, Hanhyup S; HG, Hanhyup G; HV, Hanhyup V; HA, Hanhyup A; HZ, Hanhyup Z; Ab, Arbor Acre; Cobb, Cobb broiler; Ross, Ross broiler; LO, Lohmann brown; HL, Hyline brown.

**Figure 3 animals-11-00241-f003:**
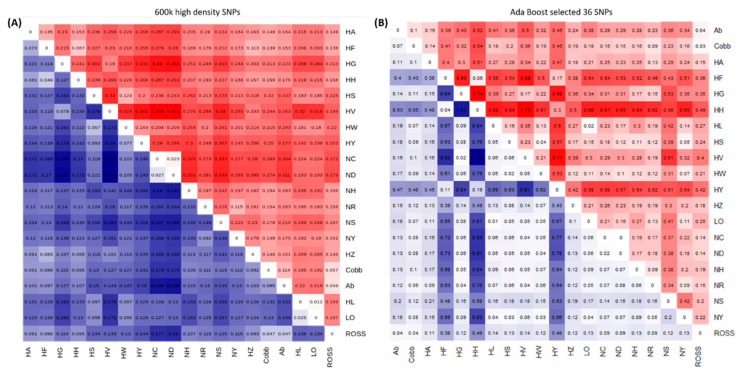
Heatmap showing the genetic distance and fixation index (Fst) results. Genetic distances are shown in blue, and Fst values are shown in red. (**A**) High-density SNPs had reasonable genetic distances and Fst values in their genetic relationships. (**B**) The selected marker combination with 36 SNPs had relatively large genetic distances between the target (case) population and other (control) chicken populations. NC, Rhode Island Red C; ND, Rhode Island Red D; NH, Cornish H; NS, Cornish S; NR, Red Korean native chicken; NY, Yellow Korean native chicken; HH, Hanhyup H; HF, Hanhyup F; HY, Hanhyup Y; HW, Hanhyup W; HS, Hanhyup S; HG, Hanhyup G; HV, Hanhyup V; HA, Hanhyup A; HZ, Hanhyup Z; Ab, Arbor Acre; Cobb, Cobb broiler; Ross, Ross broiler; LO, Lohmann brown; HL, Hyline brown.

**Figure 4 animals-11-00241-f004:**
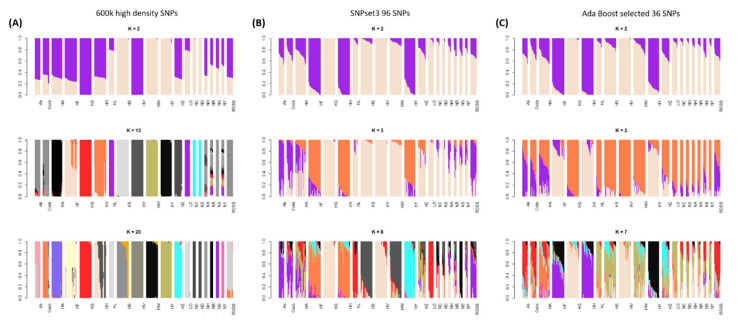
Admixture results using the data from 600 k SNPs, selected 96 SNPs, and selected 36 SNPs, to identify the genetic components of the chicken population. (**A**) The genetic component of the 20-chicken population was identified as 12 components, through cross-validation (CV) error analysis. (**B**) 50-LD pruned 96 SNPs confirmed k = 8 as optimum CV error and (**C**) 36 SNPs by feature-selection function of AdaBoost model detected two of optimum CV errors for the classification of targeted chickens. NC, Rhode Island Red C; ND, Rhode Island Red D; NH, Cornish H; NS, Cornish S; NR, Red Korean native chicken; NY, Yellow Korean native chicken; HH, Hanhyup H; HF, Hanhyup F; HY, Hanhyup Y; HW, Hanhyup W; HS, Hanhyup S; HG, Hanhyup G; HV, Hanhyup V; HA, Hanhyup A; HZ, Hanhyup Z; Ab, Arbor Acre; Cobb, Cobb broiler; Ross, Ross broiler; LO, Lohmann brown; HL, Hyline brown.

**Figure 5 animals-11-00241-f005:**
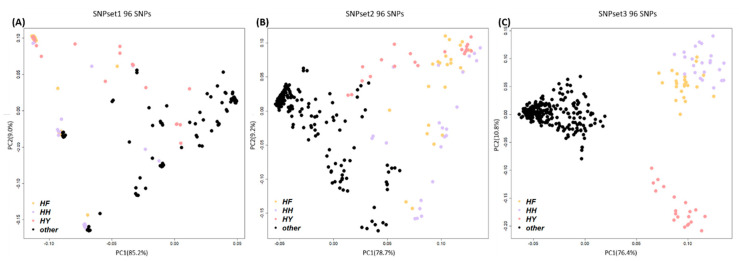
The optimal marker combination used to distinguish the case group (HH, HF, and HY) from the other chicken groups, considering population linkage disequilibrium (LD). (**A**) SNPset1 result using 96 SNP markers with high *p*-values produced in the association analysis, without considering the LD in the chicken population. (**B**) SNPset2 result using 96 SNPs that had undergone 1-LD pruning. (**C**) SNPset3 result using 96 SNPs that had undergone 50-LD pruning. The best marker combination for distinguishing between the case and control groups was SNPset3. HH, Hanhyup H; HF, Hanhyup F; HY, Hanhyup Y; other, all other chicken populations.

**Figure 6 animals-11-00241-f006:**
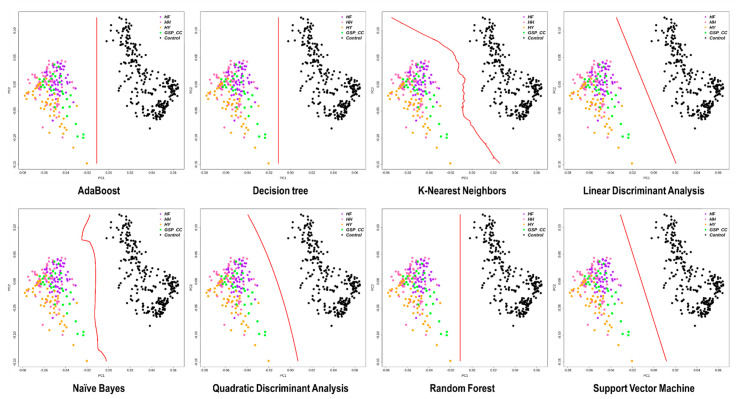
Classification results for the case and control groups, obtained by applying machine learning classification algorithms. All machine learning models could distinguish target (case) chicken populations from all other (control) chicken populations. HH, Hanhyup H; HF, Hanhyup F; HY, Hanhyup Y; GSP_CC, 3 way crossing offspring of HH, HF, and HY;Control, Other all chicken populations.

**Figure 7 animals-11-00241-f007:**
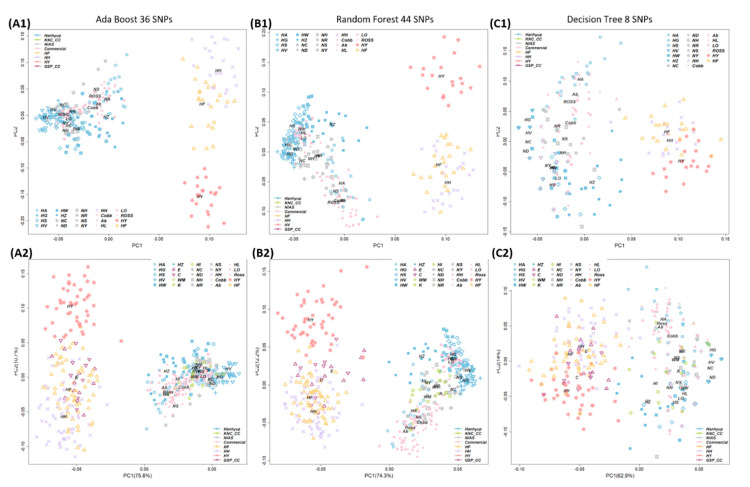
Classification results using marker combinations with the minimum number of SNPs (36, 44, and 8, respectively) selected by the feature-selection function of AdaBoost, Random Forest, and Decision Tree machine learning algorithms. (**A1**,**B1**,**C1**) Classification results using Sample Set 1 (selected markers). (**A2**,**B2**,**C2**) Classification results using Sample Set 2 (validation samples) after Sample Set 1 was used as training data. In the verification stage, the best-fitting models were the AdaBoost and Random Forest models. The minimum number of markers was set as 36 in the AdaBoost model. NC, Rhode Island Red C; ND, Rhode Island Red D; NH, Cornish H; NS, Cornish S; NR, Red Korean native chicken; NY, Yellow Korean native chicken; HH, Hanhyup H; HF, Hanhyup F; HY, Hanhyup Y; HW, Hanhyup W; HS, Hanhyup S; HG, Hanhyup G; HV, Hanhyup V; HA, Hanhyup A; HZ, Hanhyup Z; Ab, Arbor Acre; Cobb, Cobb broiler; Ross, Ross broiler; LO, Lohmann brown; HL, Hyline brown.

**Table 1 animals-11-00241-t001:** Details of the samples used in this study.

Chicken Group	Population Code	Origin of Population	Description	600 k Array (Sample Set 1)	Validation (Sample Set 2)
Government-maintained chicken (NIAS)	NC	Rhode Island Red	Imported (1960s) and locally adapted chicken population	6	
	ND			6	
	NH	Cornish		6	
	NS			6	
	NR	Red-brown Korean native chicken	Purebred Korean native chicken	6	
	NY	Yellow-brown Korean native chicken		5	
Commercial native chicken	HH	Hanhyup Farm	Founder population for new chicken breeding stock	23	36
	HF			23	36
	HY			21	26
	HW		Maintained population	23	
	HS			23	
	HG			23	
	HV			23	
	HA			20	
	HZ			15	
	1E		HH, HF, HY cross		10
	2C		HH, HF, HY cross		10
	WM_2	Woorimatdaq ver2	NIAS-developed crossed chicken population		10
	Yelim K	Yelim Farm	Private population		5
	HI	Hyunin Farm	Private population		5
Commercial broiler	Ab	Arbor Acre	Meat-type chicken	10	11
	Cobb	Cobb broiler		12	8
	Ross	Ross broiler		12	20
Commercial layer	LO	Lohmann brown	Egg-producing chicken	10	5
	HL	Hyline brown		10	
Total				283	182

NIAS, National Institute for Animal Science.

**Table 2 animals-11-00241-t002:** Accuracy of identifying the target (case) chicken population by using the minimum number of SNP markers (feature selection) selected by different machine learning models.

Classification Model	Accuracy	^1^ AUC	Precision	Sensitivity (^2^ TPR)	Specificity (^3^ TNR)
**AdaBoost: 33 selected markers**					
Decision Tree	0.995	0.996	1	0.992	1
AdaBoost	0.995	0.996	1	0.992	1
Linear ^4^ SVM	0.995	0.996	1	0.992	1
^5^ QDA	0.995	0.996	1	0.992	1
Random Forest	0.995	0.996	1	0.992	1
^6^ LDA	1	1	1	1	1
K-Nearest Neighbors	1	1	1	1	1
Naïve Bayes	0.995	0.996	1	0.992	1
**Random Forest: 44 selected markers**					
Decision Tree	0.969	0.975	1	0.949	1
AdaBoost	0.969	0.975	1	0.949	1
Linear SVM	0.990	0.992	1	0.983	1
QDA	0.995	0.996	1	0.992	1
Random Forest	0.969	0.975	1	0.949	1
LDA	1	1	1	1	1
K-Nearest Neighbors	0.984	0.987	1	0.975	1
Naïve Bayes	0.969	0.975	1	0.949	1
**Decision Tree: 8 selected markers**					
Decision Tree	0.984	0.987	1	0.975	1
AdaBoost	0.984	0.987	1	0.975	1
Linear SVM	0.964	0.970	1	0.941	1
QDA	0.974	0.979	1	0.958	1
Random Forest	0.979	0.981	0.991	0.975	0.986
LDA	0.995	0.996	1	0.992	1
K-Nearest Neighbors	0.984	0.987	1	0.975	1
Naïve Bayes	0.969	0.975	1	0.949	1

^1^ AUC, area under curve; ^2^ TPR, true-positive rate; ^3^ TNR, true negative rate; ^4^ SVM, support vector machine; ^5^ QDA, quadratic discriminant analysis; ^6^ LDA, linear discriminant analysis.

**Table 3 animals-11-00241-t003:** Classification of validation samples, using the 36 selected SNP markers.

Pop	N	AdaBoost	Random Forest	Decision Tree	Linear DiscriminantAnalysis	Naïve Bayes	Nearest Neighbor	Quadratic Discriminant Analysis
HH	36	1	1	0.972	1	1	0.972	0.972
HF	36	1	1	0.972	1	1	1	1
HY	26	0.962	0.885	0.769	0.962	1	0.962	0.923
1E	10	1	0.7	0.9	1	1	1	1
2C	10	1	1	0.8	1	1	1	0.9
Ab	11	1	1	1	1	1	1	1
Cobb	8	1	1	1	1	1	1	1
Ross	20	1	1	1	1	1	1	1
LO	5	1	1	1	1	1	1	1
WM_2	10	1	1	1	1	1	1	1
Yelim K	5	1	1	1	1	1	1	1
HI	5	1	11	1	1	1	1	1

Pop, population; N, number of predicted samples; HH, Hanhyup H; HF, Hanhyup F; HY, Hanhyup Y; 1E and 2C, three-way commercial chicken (HH, HF, and HY); Ab, Arbor Acre; Cobb, Cobb broiler; Ross, Ross broiler; LO, Lohmann brown; WM_2, Woorimatdaq ver2 commercial chicken; Yelim K, Yelim Farm commercial chicken; HI, Hyunin commercial chicken.

## Data Availability

The data presented in this study are available on request from the corresponding author. The data are not publicly available due to the agreement with funding bodies.

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
