# Peer review of "Identification of Target Chicken Populations by Machine Learning Models Using the Minimum Number of SNPs"

_animals, 2021, doi:10.3390/ani11010241_

Round 1

Reviewer 1 Report

Dear author,
Thank you for submitting your revised manuscript.
I think it was much easier to understand than the original manuscript.
I am satisfied with the responses and the edits, I am happy to accept this manuscript. Therefore, I have no further comments to make, all of my previous concerns were adequately addressed. This manuscript will be satiating the reader's interest.

Author Response

Thank you for your careful review and encouragement for this manuscript.

I attached the answer in a word file.

Reviewer 2 Report

Dear authors,

I read the revised version of your paper, and although it has improved I still think that it needs revisions. In particular, English is still poor and must be revised.

Specific comments:

L55: please be more specific, add an example of such bioactive compounds
L91-93: this sentence is still not clear: what do you want to say here? I still don't see the connection you are trying to make between machine learning models based on genomic data and F and Delta statistics, eigenvalues, PCA.
L144-149: you say you did not need to impute missing genotypes because, if I understood correctly, you removed missing data from your dataset. Still you need to write in the paper what was the missing rate in your initial SNP data panel and how many SNP loci were left after you removed SNP with missing genotypes
L192-193: what do you mean by "25 maximum decision tree coefficient-max number of sub-populations"?? As a minimum, RF has two tuning parameters, the number of trees in the forest and the number of variables (features) randomly selected in each tree.
L500-502: "This efficient search and use of SNP for the classification of breeds are thought to be used in various fields such as the traceability system of economic animal meat such as cattle, pigs and chickens, forensic science, and conservation genetics.": i) poor English (e.g. The efficient research ... is thought to be [singular]); ii) what do you mean by "economic animal meat"?; iii) please add citations for the use of SNP in traceability (e.g. Ramos et al 2011, Animal Genetics), forensic science (e.g. Ciampolini et al. 2017, BMC Research Notes), conservations genetics (e.g. Carroll et al. 2018, Evolutionary Applications; Wasser et al. 2010, Science)
L502-503: "In addition, if we could further develop our research, Biscarini et al., (2015)[46] presented a model for predicting nearly 100% of the various properties in 30 to 50 SNPs for the root vitality of sugar beet." I would rephrase to something like "... a model for predicting the root vigor class in sugar beets with nearly 100% accuracy based on a minimum set of 30-50 SNPs"
L505-507: I don't really understand this: the paper you cite is about binary classification

Figure 1: please make the caption clearer: you need to specify all the abbreviations you use in the Figure (e.g. what do you mean by ! LD and 50 LD pruning? etc.)
Table 2: it is not clear whether this accuracy was measured on the training set (set1) or on the validation set (set2). I suggest you replace specificity and sensitivity with less ambiguous terminology like true negative and true positive rates (TNR, TPR)

Author Response

Thank you for your careful review of this manuscript.
I was able to correct many mistakes. Thanks once again.
I attached the answer in a word file.

Round 2

Reviewer 2 Report

Dear authors,

I only have one residual comment: 

L153: when you say genotype error < 90% do you refer to missing rate? (i.e. remove SNP with call-rate < 90% / missing-rate > 10%?) It is also not clear what the numbers between brackets (1,126 ... 27,317) represent: number of SNPs removed?

Author Response

Comment: L153: when you say genotype error < 90% do you refer to missing rate? (i.e. remove SNP with call-rate < 90% / missing-rate > 10%?) It is also not clear what the numbers between brackets (1,126 ... 27,317) represent: number of SNPs removed?

Response: As suggested by the reviewer, it seems that this content needs to be clearly marked. we corrected it to the following content. Thank you again.

Revised manuscript (line 153-157): A total of 545,563 SNPs were obtained from the merged common SNPs from the PLINK binary data, and this result was subjected to a QC procedure, with the two main criteria of genotype error (missing rate > 10%; 1,126 SNPs removed) and minor allele frequency (<0.01; 27,317 SNPs removed) used for the selection of SNP markers in genetic diversity analyses.

This manuscript is a resubmission of an earlier submission. The following is a list of the peer review reports and author responses from that submission.

Round 1

Reviewer 1 Report

Dear Authors,

In this study, a total of 20 lines and 283 samples which consisted of Korean native chickens, commercial native chickens, and commercial broilers with layer population were used for finding the minimum number of marker combinations through the 600k high-density single nucleotide polymorphism (SNP) array. In addition, authors developed machine learning models based on the GWAS and LD analysis, which could be efficient for selecting the minimum number of SNP markers needed for KNC breed classifications.

I thought that the authors were fully analyzing 20 lines and 283 chicken samples.

If this study becomes clear, I think your research will be an important genetic resource of breed classification research.

However, I think that it is necessary to strengthen the reliability of the result by adding as much information as possible.

As a result, for the review this time, I request a major revision to the manuscript.

The reason for the major revision is that the methods and results are not clear.

Also, there is no/little information about the method and result details (Fst analysis etc.). Although result data is very detailed, method of the Fst analysis is not described enough in the methods section.

Therefore, I think your research is good, but your results are not clear enough for most readers to understand. I strongly recommend checking and correcting your manuscript by some experts or authors.

Authors need a major revision for submission.

I request a major revision to the above manuscript.

Also, there are some comments for your manuscript.

Please use this for reference as well.

1. There are few references in the Introduction section. There are many such papers, so please increase the number of citations and deepen the content of the introduction.

2. Line 108 2.1. Experimental animals and DNA extraction of the Materials and Methods

The Experimental animals and DNA extraction sections need to be separated into two sections.

3. Page 7, Line 221-242

Please explain the reason for the founder population from the results by the fixation index (Fst).

4. Please add the calculation method and description of Fst to the section of Materials and Methods.

Also, Why Fst instead of Gst ?

5. Because the seven figures and results are not clear enough to understand, it is impossible to do a qualified confirmation. 

6. I think this is probably because the founder populations of chickens vary from country to country.

Therefore, I think it would be good to add “Korean native chicken” to the title, but  I would like to know the authors' ideas.

>>Identification of (Korean native chicken) Populations by Machine Learning Models Using the Minimum Number of SNPs

7. What is the purpose of this study?

KNC breeds identification by gene makers.

or

Development of Machine Learning Models based on KNC genetics.

or

Is it both?

Therefore, it seems that the Simple Summary, Abstract, and Conclusions are not unified. Please correct it.

8. The authors, after clarifying the purpose of their manuscript, should be clearly stated in each 4.1, 4.2, and 4.3. sections in the discussion.

9. Now that the results of this research have been shown, I would like the authors to write at the end of the “Discussion” section how they will advance this research in the future.

Will it be applied to other domesticated animals such as dogs, pigs, cows and cats?

Is it possible to do that?

Please fill in specifically.

10. Finally, I recommend editing the English in your manuscript again.

Author Response

Please find enclosed the revised version of our manuscript, entitled “Identification of target chicken populations by machine learning models using the minimum number of SNPs” written by Seo et al. Based on the reviewers’ suggestions, we have dealt with the comments as the attachment

Reviewer 2 Report

Dear authors,

I read with interest your article "Identification of Chicken Populations by Machine Learning Models Using the Minimum Number of SNPs": I found it has merit, but requires a number of modifications and improvements before it can be reconsidered for publication. Please consider and address all my general and specific comments.

- the English needs to be improved
- a useful paper for your methodology and discussion is "Developing a parsimonious predictor for binary traits in sugar beet (Beta vulgaris)" (2015) where the authors presented a methodology to identify the minimum set of SNP required to correctly classify samples in two categories using SNP genotype data.

Introduction
------------
L56: what do you mean by "functionality" here?
L57: a reference may be useful for this sentence
L62: heritable genetic characteristics
L63: written in this way it seems that these markers can differentiate cattle breeds from pig breeds, while I think you want to say that your markers can differentiate between cattle breeds and between pig breeds. Please rephrase
L84: "Data mining, string analysis, and pattern analysis can be performed using various classification models.": what is the purpose of this sentence? Please remove
L88-89: "Studies of machine learning algorithms for classification have not applied them to raw data, instead using filtered data, such as F statistic, delta statistic, or principal component analysis (PCA) eigenvalues, for classification.": this sentence is not clear: who/when applied machine learning on filtered data? For what purpose? Preprocessing of data is a common step in a typical machine learning pipeline for data analysis, and preprocessing often includes data cleaning and filtering (e.g. anomaly detection, or with SNP their minor allele frequency or the call rate etc.). Using principal components (not the eigenvalues) as input data for classification is less common, unless for instance some sort of dimensionality reduction is necessary. Please clarify
L102: why do you say "unfairly distributed in the market"?

Material and Methods
--------------------
- it is not totally clear how you performed your validation: did you use only the 283 samples to identify the candidate SNP markers via association analysis (p-values), LD blocks etc. and then to develop your classifier? If so, I assume that the 182 validation samples were not used at all in this process (SNP selection and development of the classifier), and were only used at the end to measure the accuracy of classification in the validation dataset. Is it so? Please clarify. You may consider adding a diagram to illustrate your development and validation process.
- why did you select precisely 96 SNPs?
- you did not mention any imputation of the missing SNP genotypes: how did you deal with missing data in Random Forest, boosting, discriminant analysis, Naive Bayes? (in KNN it is pretty straightforward, unless you have many missing data)
- more details on the machine learning methods and models are needed: e.g. how many neighbours did you use in KNN? How many trees, and how many variables in RF? What about the learning rate in boosting? Was any tuning of the hyperparameters performed?

Results
-------
- you say that you sought the minimum number of markers required for classification, however you did not illustrate how you did this search for the minimum set of SNPs. In Figure 1 you mention feature selection, but there is nothing on feature selection in your M&M
- SVM is not even mentioned in Material and Methods: please add, with details

L221: Fst and genetic distances were not mentioned in Methods: it is not clear what you did (e.g. which genetic distances were calculated?) and why. So far this seems more a study on the genetic characterization of chicken breeds/populations, rather than a work to identify the minimum number of SNPs required for traceability (which is what you say in the title and in the Introduction of your paper)
L267: this is not correct: your alpha is 0.05, your significance threshold is alpha/m, where m is the number of tests, since you applied Bonferroni correction
L295-296: how were these 36, 44 and 8 markers identified?
L304-306: so the validation samples were added to the training sample rather than used as proper test/validation set? And you used only 96 SNPs in this case, preselected based on association study or LD: what about the 44, 36 and 8 SNPs you mentioned earlier? These were not tested? I find your experimental design confusing, you mjust make an effort to present it in a clearer way and with more important details
L306.308: "When we selected the minimum number of SNP markers using the ‘feature selection’ function for the AB, DT, and RF algorithms, discriminatory power exceeding 99% for all three models using only the 283 samples in sample set1.": this sentence does not make sense, please check and modify

Discussion
-----------
L421-434: as I already mentioned, the article "Developing a parsimonious predictor for binary traits in sugar beet (Beta vulgaris)" (2015) used machine learning approaches to identify the minimum number of SNP markers for efficient classification. In this section you should discuss the relevant literature on reducing SNP numbers for classification problems, and I am sure that also other works have beend published on this topic in plants, animals, humans (and more generally in the machine learning literature). This would be far more interesting and relevant than generic paragraphs like "Machine learning is a supervised learning approach for classifying new observations that can be used to classify bi-class or multi-class data. Machine learning can be used for voice and handwriting analysis, and document classification. In recent years, machine learning and deep learning algorithms have been used to determine phenotypic associations (e.g., in the genome, transcriptome, and methylome) in “omics” research, and to establish classification models" (L421-425)

Author Response

(The authors gave the same response as above.)
